# Short- and Long-Term Evolution of Microbial Cultures: A Thermodynamic Perspective

**DOI:** 10.3390/ijms26094187

**Published:** 2025-04-28

**Authors:** Alberto Schiraldi

**Affiliations:** Formerly at Department of Food Environment and Nutrition Sciences (DeFENS), University of Milan, 20133 Milan, Italy; alberto.schiraldi@unimi.it

**Keywords:** microbial cultures, evolution, thermodynamics

## Abstract

A thermodynamic description of cell duplication reflects Odum’s view of a feedback energy loop that sustains the transformation of the energy of substrates in the (higher quality) energy of new microbial cells with the dissipation of heat and the (lower quality) energy of catabolites. For a closed batch microbial culture, entropy increases during the whole growth and decay cycle, i.e., the production of entropy during the growth phase displays a rate proportional to the number of cell duplications per unit time, while during the decay phase, it depends on the death rate. Because of its high mobility, water is assumed to exhibit the same thermodynamic activity throughout the system. This assumption leads to the conclusion that an osmotic balance exists between cells and their surrounding medium, which, in a closed batch culture, can affect the rate and the extent of the microbial growth. Finally, the paper suggests a thermodynamic interpretation of the increase in fitness observed in a long-term evolution experiment (LTEE), based on the supposed exergy difference between the generating and generated cells in each duplication, which is also a measure of the “age” of the cells, i.e., aged cells die first. This produces microbial cultures richer in cells with enhanced duplication potential after the many thousand generations considered in an LTEE.

## 1. Introduction

A batch microbial culture is a system in which a large portion of the available substrate becomes living matter. The increase in the microbial population looks like an autocatalytic reaction, where some amount of substrate, *s*, is transformed in some amount of catabolite, *k*, and a new cell, as follows: *s* + *cell* → *k* + *2* cells. The complexity of the products makes the idea of a backward process very unlikely. Once started, the process proceeds forward until some hurdle stops it, including the exhaustion of the substrate, in the case of a closed system.

However, such a reaction-like description does not suggest the reason why the process takes place, but simply allows for a tentative energy balance, mainly based on the laws of thermochemistry [1]. According to many authors ([2,3,4,5] and the quotations therein), the process occurs because the microbial culture is a system expressing conditions far from thermodynamic equilibrium, which produces entropy as fast as possible through the formation and multiplication of organized dissipative structures aimed at this scope.

Whether or not this may explain the origin of life is still a matter of discussion and speculation beyond the scope of this paper, which begins with the evidence that living systems seem to be the only means of generating living offspring, thanks to the supply of energy and substrates obtained from external sources. The aim of the paper is the thermodynamic description of such a process via reviewing literature reports and adding some additional considerations.

A genuinely thermodynamic description of a microbial culture and its evolution must be independent of any model of the underlying biochemical mechanism, relying only on the principles of the conservation of energy and the increase in entropy. The aim of such a description is the justification of the macroscopic evidence of the changes undergone by the microbial culture viewed as a whole, namely, cells + medium, which can exchange matter and/or energy (i.e., in an open or closed system) with the surrounding environment. The macroscopic evidence reflects a short-term evolution, which corresponds to the sigmoid growth curve that fits plate count or optical density experimental data, and a long-term evolution, which considers changes in reproductive fitness, as well as phenotypic and genomic mutations that appear after a chain of successive generations.

## 2. Short-Term Evolution of a Microbial Culture

Every microbial culture behaves like a factory in which substrates become living organisms and catabolites. This transformation is a peculiar property of the living cells, which trigger cell duplication and are themselves accelerators of its evolution, as in an autocatalytic process. However, the growth rate of any microbial culture is not constant, but increases a maximum level and then declines [6]. This trend in the population density, *N*, is the growth curve [7], currently represented in a semi-log plot, where it appears with a sigmoid shape. A number of functions can reproduce such a shape, each reflecting a specific model that assigns pivotal meaning to few parameters assumed to represent the underlying growth mechanism.

The literature is replete with papers that propose logistic, kinetic, and metabolic models aimed at reproducing macroscopic evidence of the growth curve [8,9,10]. Some of these works deal with the cellular metabolism underlying microbial growth [11,12,13,14,15] and in some case, report typical expressions of the thermodynamics of irreversible processes, coupling the growth rate with the overall energy balance of the metabolic processes [16,17,18].

However, the large uncertainty of the experimental data (plate counts or OD records, as well as the enthalpy of the formation of biopolymers), in many cases, makes the selection of a preferable model suspicious [15]. The main reason for such a failure is that the use of only a few parameters makes it impossible to reproduce the effects of the myriad of biochemical processes that underlie the growth of a microbial population [16,17,18]. This suggests that the sigmoid trend of the growth curve is independent of the peculiarities that characterize the various microbial species at the molecular level.

This led to the development of a standard, simple system, namely, a virtual microbial culture described using only a few macroscopic variables, e.g., the density of the microbial population, *N*, and the duplication rate, represented with suitable simple functions of time, which allow for an empirical reproduction of the sigmoid growth trend recorded from a real culture. In spite of its semi-empirical character, this model allowed the gathering all the sigmoid growth curves of pro- and eukaryotic microbes, in any medium and at any temperature, in a single master plot of reduced variables [19,20,21,22]. For the sake of providing an easy check, Appendix A of the present paper summarizes the main formal expressions of this model, since these are of help to show some relationships between the growth curve of a microbial culture and selected thermodynamic properties, including entropy production, water activity, and osmotic pressure.

### 2.1. A Classic Thermodynamic Balance

The chemical transformation that fuels the growth process takes place within the cell. This means that substrates, catabolites, and water must cross the cell wall through some energy-consuming process and that the dry mass (DNA, proteins, ATP, etc.) of the new cell must be enveloped and separated from the mother cell through the extension of the exchange surface with the surrounding medium, once again with consumption of energy. The generating cell “uses” its available free energy (E**x**ergy) to sustain the above processes: it is not a simple “catalyst”, but the main promoter of the duplication. The drop in exergy related to the degradation of substrate to catabolites, −Δ_chem_*G*, overbalances this loss, allowing the formation of a new cell and the release of heat toward the external environment. The generating cell is akin to a system that self-organizes by “building” a subsystem (the new cell) that will increase the surface and the rate of the uptake-and-release of matter and energy. The exchange with the surrounding medium is indeed a main bottleneck in the duplication process.

The microbial population, as a whole, behaves as a feedback reservoir of extra energy that sustains the overall growth process, its overall exergy (namely, minus Gibbs free energy) increasing with the increase in the population.

The schematic illustration of the cell duplication (Figure 1) reflects the Odum’s view of feedback loops that allow for the transformation of the energy of the substrate in the “higher quality” energy of new cells and the “lower quality” energy of catabolites and heat. If the system is open, the flow of energy corresponds to the maximum power principle, with an increase in E**m**ergy at each intermediate step [23]. If the system is closed, the evolution will reach a maximum rate, but it soon adopts a slower pace and eventually stops (see below).

From a thermodynamic perspective, a closed batch microbial culture is a system that burns its exergy, releasing part of it toward the external environment, with an increase in the overall (system + environment) entropy. The duplication enthalpy, Δ_d_*H*, includes the enthalpy drop for the chemical transformation of the substrates in the catabolites, Δ_chem_*H* = (*H*_s_ − *H*_k_), and the enthalpy of formation of a new cell, *H*_c_. Since the latter remains within the system, the heat released toward the external environment corresponds to Δ_chem_*H* < 0. Since *H*_c_ is the sum of the formation enthalpies of the compounds (multiplied by the respective mass) of the dry matter of a single cell, which are all negative, then *H*_c_ < 0. This thermochemical evaluation is a matter thoroughly treated by some authors [1]. Therefore, the enthalpy drop of a single duplication is Δ_d_*H* = (Δ_chem_*H* + *H*_c_) < 0. One has to take into account that *S*_c_ > 0, being the sum of the entropies of the compounds of the cell dry matter. This means that *G*_c_ = (*H*_c_ − *TS*_c_) < 0. Since Δ_chem_*G* = (*G*_k_ − *G*_s_) < 0, the Gibbs free energy balance for a single duplication is negative, as for any spontaneous process, as shown below:Δ_d_*G* = *G*_c_ + Δ_chem_*G* < 0,(1)

As for the entropy of duplication, Δ_d_*S* = *S*_c_ + (*S*_k_ − *S*_s_) = (*S*_c_ + Δ_chem_*S*), Δ_chem_*S* can be either positive or negative. If it is positive, the overall process is both enthalpy- and entropy-driven, while if it is negative, and |Δ_chem_*S*| > *S*_c_, the overall process is enthalpy-driven. The discussion of several cases, including the rare endothermic growth of *Methanosarcina barkeri* [24], appears in an interesting paper by von Stockar et al. [25]. It is sufficient here to note that the irreversibility of the overall process mainly depends on the negative sign of Δ_d_*G*. This condition includes the possibility that the transformation of substrate in living matter can occur with either a decrease or increase in entropy, in spite of the fact that the cell represents an “organized” subsystem, which means that it is a low entropy structure. As the microbial population grows, its overall entropy, *S*_population_ = (*N* − *N*_0_) *S*_c_, increases, while the surrounding medium that loses substrate and receives catabolites can undergo an increase or a decrease in entropy depending on the sign and the absolute value of Δ_chem_*S*.

A similar entropy balance also applies to the decay phase. Once again, a reaction-like representation of the decay of the microbial population can be of help, i.e., *cell* → *catabolites*. In this case, the process is endothermic, Δ_δ_*H* = (*H*_k_ − *H*_c_) > 0, but implies a barely detectable inbound energy flow from the external environment [26]. Δ_δ_*S* = (*S*_k_ − *S*_c_) > 0, as the relevant catabolites are the debris of the “organized” cellular mass. The spontaneous nature of the process imposes the free energy drop of Δ_δ_*G* = (*G*_k_ − *G*_c_) < 0. This means that *T*Δ_δ_*S* > Δ_δ_*H* > 0.

The above picture is “static”, in that it does not take into account the rate of the considered processes. This compels the use the relationships of the thermodynamics of irreversible processes to describe the evolution of the microbial culture. In particular, the “wasted” energy is larger than |Δ_chem_*H*|; thus, an extra term, *TS*_i_ > 0, which increases with the rate of the processes considered, must be added. Prigogine’s approach predicts the time derivative $\frac{{dS}_{i}}{dt}=\dot{S}$*_i_* to be the product of the reaction affinity and the reaction rate (duplication rate of 1/*τ* in the present case; see Appendix A),(2)$${\dot{S}}_{i}=\frac{\left(-\Delta_{d}G/T\right)}{\tau }\geq 0$$

This expression suggests that a loss of exergy takes place because of the irreversibility of the process and that such a loss corresponds to an increase in the entropy of the system with the following rate:(3)$$T{\dot{S}}_{i}=\frac{\left(-\Delta_{d}G\right)}{\tau }=-{\dot{\epsilon }}_{i}$$

Splitting Δ_d_*G* according to Equation (1) yields(4)$$\frac{\left(-\Delta_{d}G\right)}{\tau }=\frac{{-\Delta }_{d}H{+T\Delta }_{d}S}{\tau }=\frac{{-\Delta }_{chem}H}{\tau }+\frac{{-H}_{c}{+T\Delta }_{d}S}{\tau }$$

As mentioned above, Δ_chem_*H* is the heat released toward the external environment, namely, a loss of exergy related to the metabolism of the substrate. The corresponding rate is the outbound heat flow per each duplication,(5)$$\dot{q}=\frac{\Delta_{chem}H}{\tau }<0$$

The remaining term, (−*H*_c_+*T*Δ_d_*S*) = (−*G*_c_ +*T*Δ_chem_*S*) = *ε*_m_, corresponds to the metabolic exergy transferred to the products, namely, the new cell and the catabolites. Its rate is ${\dot{\varepsilon }}_{m}=\frac{\varepsilon_{m}}{\tau }$.

One can therefore rewrite Equation (4) as $-{\dot{\varepsilon }}_{i}=(-\dot{q}+{\dot{\varepsilon }}_{m})$ or, more suitably,(6)$$\dot{q}={(\dot{\varepsilon }}_{m}+{\dot{\varepsilon }}_{i})<0$$

Multiplying the terms of Equation (7) by (*N* − *N*_0_), namely, the increase in the microbial population, yields the overall exothermic heat flow released by the microbial culture, $\dot{Q}$. Equation (7) can be of some interest for the treatment of experimental $\dot{Q}$ data collected via isothermal calorimetry that allows for a direct determination of $\dot{Q}$ as a continuous record throughout the growth process [26,27]. To this aim, the trace must be viewed as the result of two contributions,(7)$$\dot{Q}=\frac{\left({N-N}_{0}\right)}{\tau }\varepsilon_{m}+\left({N-N}_{0}\right){\dot{\varepsilon }}_{i}\;$$
leaving the assessment of the best values for *ε*_m_ and ${\dot{\varepsilon }}_{i}$, which are treated as constant parameters, to a fitting routine. This approach requires the use of functions for *N*(*t*) and *τ*(*t*), namely, a suitable model to describe the growth curve. Figure 2 reports the case of a culture of *Lactobacillus helveticus*, for which the model of the virtual culture seemed suitable.

For an open system, one of the main issues is the assessment of how far the system is from an equilibrium or a steady state. For a closed (no exchange of matter with the external environment) batch microbial culture, there is no real steady state, since the pace of the process depends on the progress of overall growth experienced by the microbial population [20,21], which implies a decrease in E**x**ergy to sustain the process.

At the onset of the growth, the overall exergy of the substrate is the highest, while that of the microbial population is the lowest, mainly because of the smallest overall exchange surface between the cells and the surrounding medium. The evolution rate is very low, but soon increases toward the maximum entropy production rate [4,5]. In our case, this means the need to search the following:(8)$$\ddot{S}=\frac{\left(\Delta_{d}G/T\right)}{\tau^{2}}\dot{\tau }=0$$

A straightforward algebra equation, based on the relationships of the model for the virtual batch culture (see Appendix A), shows that the time derivative of (1/*τ*) experiences a maximum rate in the middle of the exponential phase of the growth curve (Figure 3).

Once this condition is reached, the system tends to continue the achieved evolution pace for a while, corresponding to the so-called exponential growth phase, often dubbed “balanced growth” [6], i.e., when all the metabolic processes heading toward cell duplication show the same pace. Such a condition should imply the lowest “waste” of the exergy (of the medium).

In a closed system (no exchange of matter with the external environment), the very fast growth phase is followed by a decrease in the duplication rate. One reason for this behavior is the depletion of the available substrate. Another reason deals with the partition of water between the extra- and intracellular compartments, which imposes an upper limit to the number of duplication steps and to the maximum density of the microbial population. The system tends to a condition of rest, i.e., $\dot{N}\sim 0$, that resembles thermodynamic equilibrium. The entropy production rate tends toward minimum [2], and eventually, in a closed system, both *S*_i_ and $\dot{S_{i}}$ tend to zero, which corresponds to no energy flow between the system and the external environment.

The fact that $\dot{S_{i}}$ must experience a maximum rate and eventually drop to zero is consistent with the sigmoid shape of the growth curve observed for every microbial batch culture (either pro- or eukaryotic), no matter the underlying biochemical and genomic processes. In particular, (1/*τ*) is expected to be a function of the time, *θ*, and to depend on the size of the microbial population, *N*(*θ*), no matter the specific duplication mechanism, and to hold for every microbial culture. Indeed, this was the reason for the choice of the model of the virtual culture summarized in Appendix A of this paper.

In an open chemostatic culture, $\dot{N}$ is constant, depending on the substrate supply rate, and $\ddot{S}$ = 0 and $\dot{S_{i}}$ tend to a minimum rate, which would reflect the “dissipative adaptation” behavior described by England [28].

### 2.2. The Gibbs Free Energy Trend in a Virtual Batch Culture

The evolution of a batch microbial culture is spontaneous and therefore, occurs with decrease in its Gibbs energy, *G*, throughout the growth and decay cycle.

The virtual microbial culture used to develop our model (see Appendix A) is of help in describing such a trend. For the sake of clarity, one can split the overall continuous growth and decay decrease in the Gibbs energy of the microbial culture into two terms, *G_γ_* and *G_δ_*.

Growth(9)$$G_{\gamma }=G_{{N=N}_{0}}+\int_{0}^{\xi }\left(\frac{dG}{d\xi }\right)d\xi$$
where *ξ* = *ξ*(*θ*) = log_2_(*N*/*N*_0_)/*β*, *β* being the maximum allowed number of synchronic duplication steps and *θ* the time referred to the origin *θ* = 0 for *N* = 1 (see Appendix A).

Since $\left(G_{\gamma }-G_{{N=N}_{0}}\right)$ ≤ 0, then (d*G*/d*ξ*) < 0. Assuming that each cell duplication would contribute to the decrease in *G* by the same Δ_d_*G* < 0 and taking into account that the overall number of cell duplications is *n* = *N*_0_2*^ξ^*^(^*^β−^*^1)^, then(10)$$\left(\frac{dG}{d\xi }\right)=\Delta_{d}G\left(dn/d\xi \right)=\Delta_{d}G\;n\;{log}_{e}\left(2\right)\;(\beta -1)$$
and(11)$${(G}_{\gamma }-G_{{N=N}_{0}})=\Delta_{d}G{log}_{e}\left(2\right)(\beta -1)\int_{0}^{\xi }nd\xi =\Delta_{d}GN_{0}\;\left[2^{\xi (\beta -1)}-1\right]$$

Appendix A reports the explicit expression of *ξ*(*θ*).

Decay(12)$${(G}_{\delta }-G_{{N=N}_{0}})=\Delta_{d}GN_{0}\;\left[2^{\left(\beta -1\right)}-1\right]exp\left[-{\left(\theta -\theta_{s}\right)}^{2}/\delta \right]\mathrm{for}\;\theta \;\geq \;\theta_{s}$$where *θ*_s_ is the onset of the decay trend (see Appendix A). Figure 4 shows the corresponding trends, scaled from the onset of the duplication *θ*_0_.

The plot reported in the above figure corresponds to the 3D representation shown in Figure 5, where the growth and decay cycle is indicated by the trace on the horizontal plane.

### 2.3. The Osmotic Balance

In a closed batch culture, the matter exchanged through the cell walls modifies the compositions of the medium (potentially, the cell cytoplasm as well) and affects both the rate and the extent of the growth of the microbial population, as each duplication step occurs within a different medium. Consistently, the Gibbs free energy per unit mass of the medium, *G*_m_, and the cells, *G*_c_, change during a growth run.

Apparently, a self-adaptation mechanism is at play within the cells, which do not seem to undergo detectable modifications, while no such action takes place in the medium. Fortunately, the medium is much simpler to describe, as one can treat it as an aqueous solution and use the solvent as a reference component, gathering all the other solutes in a single virtual component, with *x*_m_ = (1 − *x*_W_) molar fraction.

The mass balance is derived as follows:*M* = *M*_w,m_ + *M*_w,c_ + *M*_c_ + *M*_m_ = constant(13)
where the subscripts stand for water in the medium, the water in the cells, the dry matter of the cells, and the dry mass of the medium, respectively. *M*_m_ is the sum of dry mass of the substrate, *M*_s_, the catabolites, *M*_k_, and the extra solutes of small molecular mass that do not undergo chemical changes. The overall mass of the catabolites is *M*_k_ = [*M*_s0_ − *m*_c_(N − N_0_)], *m*_c_ being the dry mass of a single cell.

The transfer of water occurs much faster than any other process within the system and depends on transient gradients of water activity, *a*_W_, between the next neighboring regions. This is why one may safely assume that the value of *a*_W_ in the cells, (*a*_W_)_c_, is substantially the same as that in the medium, (*a*_W_)_m_, being the result of an equilibrium condition attained at any step of the growth run. This assumption was experimentally [29] supported by an additional report [30] about the drop of osmotic pressure across the microbial cell wall, which is rather small (about 1000 Pa), and is barely adequate to sustain the inbound flow of solvent required for the cellular metabolism. The possibility that a microbial population may host regions with different *a*_W_ under unique situations, where physical hurdles hinder the molecular migration, is the subject of a separate paper [31] and is not considered in the present research. The known relationship between *a*_W_ and osmotic pressure, *Π*, is(14)$$\Pi \frac{{\bar{V}}_{W}}{RT}={-log}_{e}\left(a_{W}\right)\;\sim \;{-log}_{e}\left({1-x}_{m}\right)\;\sim \;x_{m}$$
where ${\bar{V}}_{W}$, *R* and *T* stand for the partial molar volume of water, the gas constant, and absolute temperature, respectively, and *x*_m_ is the molar fraction of a virtual solute that gathers all the dissolved compounds. The usual *a*_W_ values of microbial cultures range above 0.950, with variations in the third decimal figure producing detectable changes in both the rate and the extent of the growth curve [32]. This makes Equation (9) a rather likely approximation. Another safe approximations (since *M*_w,m_ >> *M*_m_) is(15)$${x_{m}\sim X}_{m}=\frac{M_{m}}{M_{m}+M_{w,m}}\sim \frac{M_{m}}{M_{w,m}}\propto \Pi$$
where *X*_m_ stands for the mass fraction of the solutes in the medium.

If *Π* is buffered and *a*_W_ kept above 0.950 (e.g., adding various amounts of NaCl [32] to the medium), namely, d*Π* = 0, one can easily verify that the function *X*_m_ = *X*_m_(*N*) is constant when(16)$$\frac{M_{m,0}}{M_{w,m,0}}=\frac{m_{c}}{m_{w}}$$
where *m*_W_ is the amount of water within a single cell.

Equation (17) reflects a kind of osmotic equilibrium between the medium and the cells, which is at play since the onset of the duplication process (subscript “0” added in the above symbols). A large *M*_W,m,0_ corresponds to large values of *a*_W_ and *m*_W_. When applied to the cytoplasm of the cells, this statement equates to “the larger the water activity, the smaller the viscosity”, because of the polymeric nature of most dry matter ([31] and the quotations therein). This means that the rate of every metabolic process within the cells will increase with increasing *a*_W_. This conclusion is in line with the experimental evidence reported by Mellefont et al. [32], who also observed that a lower *a*_W_ was accompanied by a longer lag phase and a smaller growth rate and extent. This correlation between growth extent, growth rate, and duration of the pre-growth latency is a main issue in the virtual batch culture model (see Appendix A).

If d*Π* ≠ 0, then the growth of the cell population implies an increase in *X*_m_(*N*), mainly because of the transfer of water into the cells. The value of *a*_W_ decreases, while the value of *Π* increases. When the mass of the available water in the medium, *M*_w_, corresponds to *a*_W_ ~ 0.8, practically no growth takes place for many microbial species. This threshold reflects an old experimental evidence, which reflects a simple interpretation. Low *a*_W_ corresponds to low molecular mobility [31], and one can recognize a “critical water activity”, *a*_WC_, (Ref. [33] and the quotations therein) for which the mobility becomes practically naught as the system approaches its glass transition or sol/gel threshold. The value of *a*_WC_ depends on the nature of the solutes (e.g., polymers within the cell and compounds of small molecular mass in the medium), being larger for the cellular cytoplasm, which looks like a gel [31], than for the outer medium. Molecular mobility vanishes first within the cells, and any metabolic reaction, including the uptake of substrate, therefore ceases, while the outer medium still displays moderate viscosity.

## 3. Long-Term Evolution of a Microbial Culture

The microbial cultures considered in most studies reported in the literature are the result of refreshing samples from stored collections at rest. This practice allows for the strict control of strains and hopefully, a correct comparison of data by different investigators. Therefore, any growth curve determined starting from such samples is a short-term evolution of the same microbial generation stored in the “official” collection.

These investigations are not suitable to describe the long-term evolution of a given microbial species, which undergoes genomic changes that could require centuries or millennia to appear in the real world, where a colony of microorganisms moves from one host to another and restarts its growth in new mediums and under different environmental conditions. Such evolution of microbial strains is of interest in phylogeny studies [34] and above all, in investigations regarding their adaptation to adverse environmental conditions, as in the case of the abuse of antibiotic drugs in feeding poultry, cattle, and crops for the food industry, and of long-term or unwise pharmacological prescriptions.

A suitable experimental approach is the long-term evolution experiment (LTEE) method, like the one undertaken by Lenski [35,36] in the last decade of the previous century using a culture of Escherichia coli, which has currently surpassed several thousand growth and refresh cycles in the same medium and from the same *N*_0_ level. The experiment aims to preserve the memory of the previous history of the culture, allowing for comparison between cultures several generations apart from one another. A major outcome of this research is the evidence of the increased reproductive fitness, defined as the growth progress in a day, with respect to the starting ancestor culture. This increase displays an asymptotic trend, just as in the evolution of metastable systems not too far from thermodynamic equilibrium [35,36], and also implies some genomic epistasis [37].

An interpretation of such evidence is of interest and is achievable using the model of the virtual batch culture described in Appendix A, although with a basic adjustment, i.e., cell duplication is not a process of perfect self-cloning. Some tiny difference must exist between the generating and generated cell. Not knowing the reason or the nature of this difference, one must adopt a blind approach, with no reference to any underlying event at the molecular level, as in a thermodynamic description.

According to Annila [38,39], exergy drops sustain any change in living organisms. If a difference exists between cells several generations apart from one another, then it may correspond to a small gap of exergy, |δ*G*_c_| ≪ |G_c_|. Such δ*G*_c_ would not affect the phenotypic or genotypic properties of the microbial population as they appear in a standard investigation, although producing some evidence only after a series of a large number of dilution-and-refresh runs in an LTEE. For example, Lenski reports a 1.8 ratio between reproductive fitness values after 50 thousand generations considered in his LTEE [40].

In a real microbial culture, such a δ*G*_c_ gain of exergy would concern only a fraction of the population. However, these empowered cells will soon outnumber the others. The model of the virtual culture indeed accounts for this effect, inasmuch as each quantity is actually an average over the entire microbial population. This means that, for the sake of consistency, the model assumes that |δ*G*_c_| (either positive or negative) should enter into the thermodynamic balance of each single duplication step of every cell. This assumption leads to a phenomenological description of the behavior of real microbial cultures using simple semi empirical functions (see Appendix A).

From an anthropic perspective, one can imagine that the generated cell gains a surplus of exergy, at the expenses of the generating cell. This view is consistent with the adopted model of the virtual culture that accounts for the age differences, the oldest cells being the first to die [21]. Age here stands for the duplication steps experienced and implies a loss of performance, namely, reduced exergy, in line with the findings of Ping Wang et al. [41], who claimed that “…death of E. coli is the result of accumulated damage”. This would occur at every duplication step during the growth phase. The neat Δ_d_*G* balance would be the same as that in Equation (1). As the cell enthalpy likely remains unchanged, δ*G*_c_ would concern just the entropy of the involved couple. A gain of exergy would correspond to a decrease in entropy and a larger reproductive efficiency.

In previous works, the algebraic expressions of the model related to the virtual batch culture [21,22] led to the realization that the maximum number of synchronic duplication steps of the virtual culture, *β*, is proportional to the product *μθ*_0_ (*μ* and *θ*_0_ standing for maximum specific growth rate and duration of the pre-duplication latency, respectively (See Appendix A). This suggested that the value of *β* could be a reasonable proxy of the reproductive fitness of the microbial culture, intended as the combination of promptness to grow (small *θ*_0_) and quickness to duplicate (large *μ*), and could replace the Lenski’s choice (growth progress in a day [40]).

In Lenski’s LTEE, each cycle starts from the same population density, *N*_0_, reconstructed by diluting a previous population at the end of its growth with fresh medium (there is no decay phase). Applying the model of the virtual batch culture, the cells of the first starting population are assumed to have the same exergy, |*G*_c_|. At the end of the first LTEE growth run (i.e., after *β*_1_ duplication steps), 1/2 of the microbial population includes cells with |*G*_c_ + δ*G*_c_| exergy. The other half accounts for fractions of 1/8 and 1/4 of the whole population that exhibit |*G*_c_ − δ*G*_c_| and |*G*_c_| exergy, respectively, and smaller fractions with even lower exergy, including the starting *N*_0_ cells that show |*G*_c_ − *β*_1_ δ*G*_c_| exergy. The less efficient cells will decay first, while those with |*G*_c_ + δ*G*_c_| exergy will survive for a longer time. Practically undetectable after a single growth step, this selection would become substantial after several thousand steps.

For the model of the virtual culture, any starting population reconstructed by dilution of the preceding one at the end of a growth run would host cells with different exergies. Half of them would exhibit a larger exergy than the other half. The overall exergy of the starting population of the new cycle would be the same as before, but its partition would change. Half of the population should be able to sustain a little larger number of duplication steps (increased *β* according to our model) than the rest of the population. This is not the case for real cultures, where the cells with the lowest exergy are not able to sustain duplication and die. With the memory of its previous history represented by the extra exergy gained after a large number of growth and refresh cycles, the microbial population will host cells with substantially larger exergy and lower entropy than those in the ancestor culture. The surviving cells then form a culture with improved reproductive fitness, which might also imply some genomic mutation and/or epistasis. However, the increase in *β* displays limits imposed by the peculiarities of the system, such as those related to the osmotic-like balance (see above) and to the heat transfer through the boundary region between the system and the external environment, since heat dissipation increases with increasing *β* because of the larger consumption of the substrate in a growth run. These physical constraints could be responsible for the asymptotic trend of the fitness increase described by Lenski and Bravenstock [35,36,37]. Figure 6 sketches the long-term evolution mimicked by an LTEE.

This interpretation also applies to microbial populations cultured in adverse mediums, where the best performing strains survive, developing some extra resilience. In this respect, one may envisage an ad hoc experimental design of chemostatic cultures, aimed at improving the Lenski’approach for an LTEE.

In Lensk’s LTEE, each cycle starts from the same population density, *N*_0_, reconstructed by diluting the population at the end of the growth with fresh medium (there is no decay phase). Could one separate evolved cells, *N*_E_, from the rest of the living matter, the increase in fitness would appear with better evidence. One way to achieve this scope is to allow a partial decay of the microbial culture. As mentioned above, the decay mainly depends on the age of the cells and therefore, affects the oldest ones first. Evolved cells should have a longer life span and therefore, survive when the rest of the microbial population undergoes death. Sampling the culture during its decay phase and selecting viable, *N*_S_, from dead, *N*_D_, cells would allow for the preparation of a culture that hosts a larger fraction of evolved cells, *N*_E_. This procedure would accelerate the evolution progress seen during the LTEE. Were it possible to continuously remove dead cells and catabolites, one could envisage a chemostatic experiment, according to the sketched design reported in Figure 7. The parameters to determine include the supply rate of the substrate and the removal rate of the catabolites, in order to keep the composition of the medium constant. For determining the heat flows, the whole equipment should reside in a suitable calorimeter. The increase in reproductive fitness would require a larger rate of supply of the substrate and larger heat flows (mainly outbound) between the chemostat and the external environment.

## 4. Conclusions

The thermodynamic description of the evolution of a microbial culture explains the macroscopic evidence of the growth curve, without any reference to the molecular level. In particular, the sigmoid shape of the growth curve observed for every batch microbial culture depends on the constraints imposed by the laws of the thermodynamics of irreversible processes.

A simple semi-empirical model, reported in previous works and summarized in Appendix A of this paper, complies with these constraints and allows for a reliable description of the macroscopic growth and decay exergy trend of any culture of duplicating microbes.

## Figures and Tables

**Figure 1 ijms-26-04187-f001:**
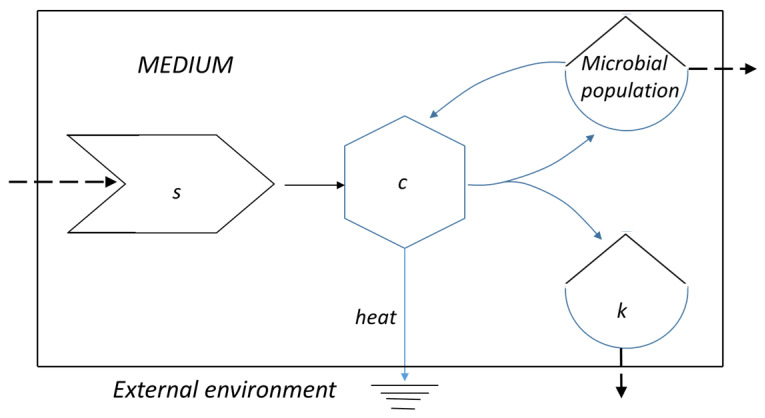
The Odum’s feedback energy supply provided by the microbial population (*c*) sustains the transformation of the substrate, *s*, in a new cell, which joins the microbial population, and the catabolites, *k*, while some amount of heat flows toward the external environment. The dashed arrows reflect the case of an open system.

**Figure 2 ijms-26-04187-f002:**
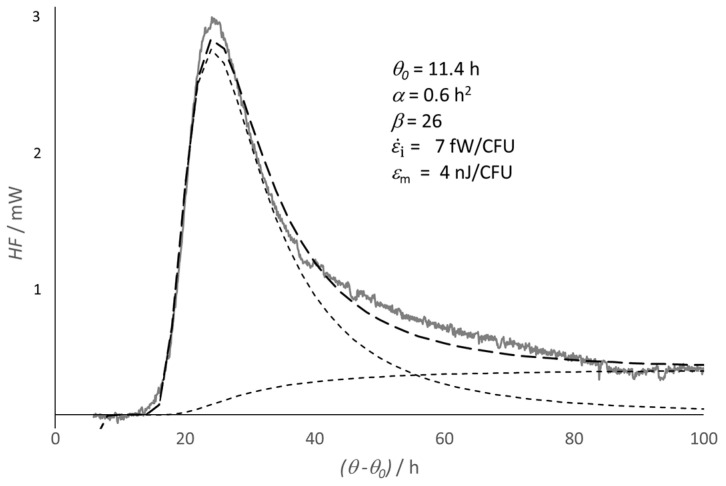
Isothermal calorimetry exothermic heat flow trace from a culture of *Lactobacillus helveticus* at 37 °C. The fitting trend (heavy dashed line) is the sum of two contributions (light dashed lines), i.e., the irreversibility, ${\dot{\varepsilon }}_{i}$, (sigmoid light trend), and the metabolic process, *ε*_m_ (dashed light peak), respectively. The time is scaled from the onset of the duplication, *θ* = *θ*_0_; *α* and *β* are the fitting parameters of the model (see Appendix A). IC trace and experimental details were reported in Ref. [27].

**Figure 3 ijms-26-04187-f003:**
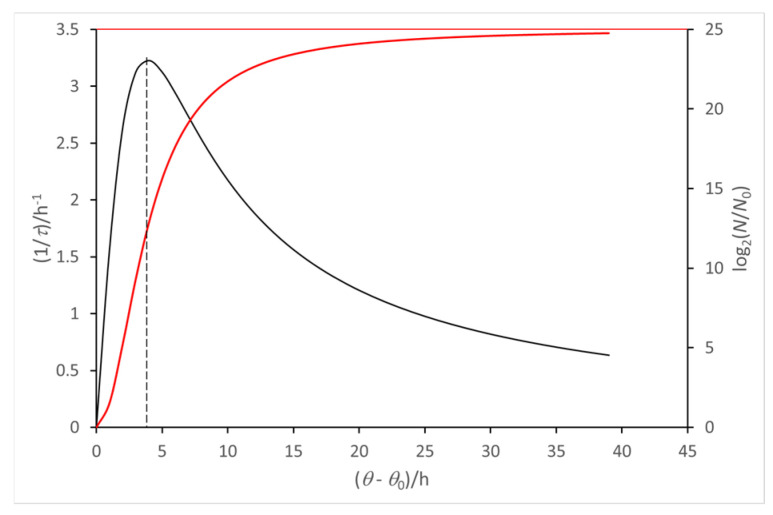
The maximum rate of entropy production coincides with the maximum duplication rate, 1/*τ*. This condition occurs in the middle of the exponential phase of the growth curve (red line). The time is scaled from the onset of duplication, *θ* = *θ*_0_ (see Appendix A).

**Figure 4 ijms-26-04187-f004:**
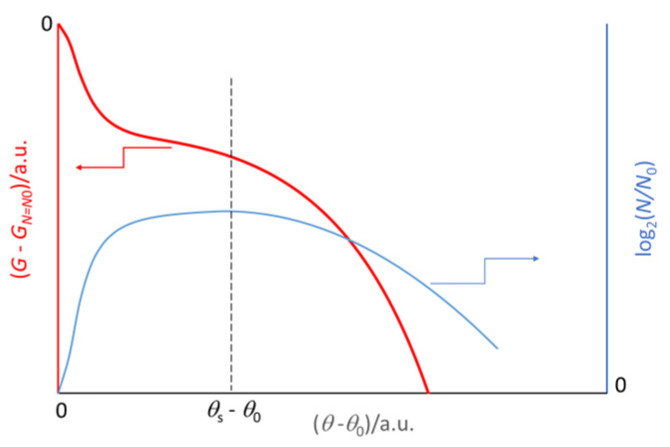
Decreasing trend in the Gibbs energy (red line) of a closed batch virtual culture that can exchange energy, but not matter, with the surrounding environment. The blue line corresponds to the growth and decay trend of the microbial population.

**Figure 5 ijms-26-04187-f005:**
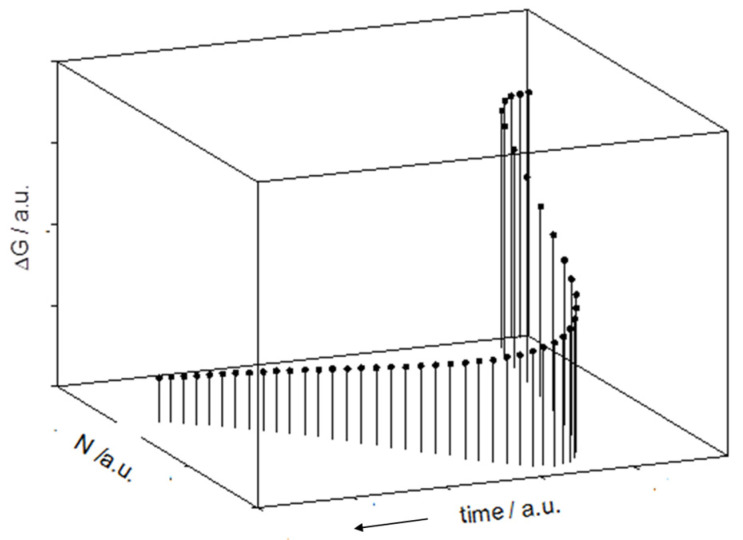
Expected decrease in the Gibbs energy during the growth and decay of a microbial culture. The projection on the horizontal plane reflects the growth and decay trend of the microbial population.

**Figure 6 ijms-26-04187-f006:**
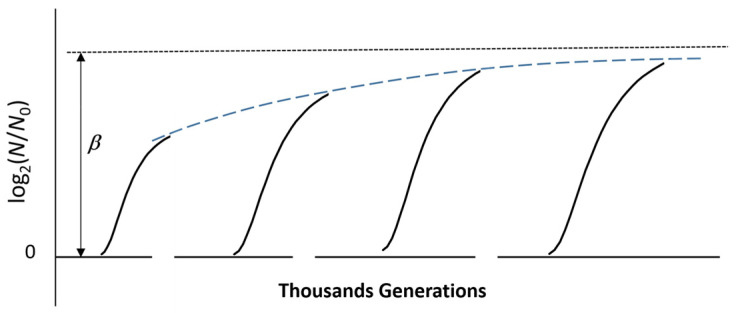
Expected succession of LTEE growth and decay cycles. Because of the increase in fitness, each growth and decay cycle implies a larger population density of better performing newborn cells. The upper limit of the reproductive fitness, *β*, depends on the osmotic balance within the system (cells + medium) and on the heat transfer through the boundary region between the system and the external environment.

**Figure 7 ijms-26-04187-f007:**
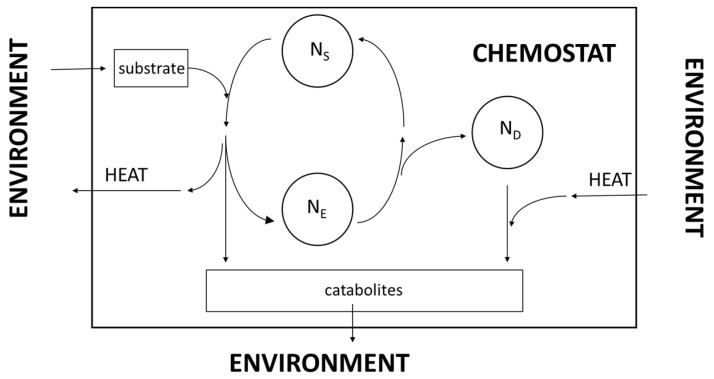
Idealized chemostat aimed at accelerating the detection of reproductive fitness. The supply rate of the substrate and the removal rate of the catabolites should be monitored to keep the composition of the medium constant. To determine the heat flows, the whole equipment should reside within a suitable calorimeter. *N*_E_, *N*_S_ and *N*_D_ stand for evolved, surviving and dead cells, respectively.

## Data Availability

Data available on request.

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
