# Peer review of "Short- and Long-Term Evolution of Microbial Cultures: A Thermodynamic Perspective"

_ijms, 2025, doi:10.3390/ijms26094187_

Round 1

Reviewer 1 Report

Comments and Suggestions for Authors

This study explores the short-term and long-term evolution of microbial cultures from a thermodynamic perspective. It presents a semi-empirical model based on the thermodynamics of irreversible processes, explaining the sigmoidal growth curve and linking the increased fitness observed in long-term evolution experiments (LTEE) to differences in exergy. This approach is somewhat novel in microbiology. However, the following revisions are suggested before reconsideration.

  1. While the study provides a thermodynamic explanation, it lacks a detailed comparison between model predictions and experimental data. For instance, validating the model's accuracy by measuring microbial growth curves and entropy production rates under various conditions would be beneficial.
  2. The discussion on the increased fitness observed in LTEE in the “Long-term Evolution” section is rather brief. A more detailed explanation of how the model accounts for this phenomenon is warranted.
  3. The study would be strengthened by including specific experimental designs to directly verify the model's predictions.
  4. The limitations of the model are not adequately addressed. For example, the assumption of uniform water activity throughout the system may not hold in complex microbial cultures. A thorough analysis of the model's limitations and suggestions for future improvements should be included in the discussion.
  5. The quality of Figures 2 and 6 is subpar, and they lack sufficient captions. Enhancing the clarity of these figures and providing detailed captions would improve the manuscript.

Author Response

  1. While the study provides a thermodynamic explanation, it lacks a detailed comparison between model predictions and experimental data. For instance, validating the model's accuracy by measuring microbial growth curves and entropy production rates under various conditions would be beneficial.
  2. The discussion on the increased fitness observed in LTEE in the “Long-term Evolution” section is rather brief. A more detailed explanation of how the model accounts for this phenomenon is warranted.
  3. The study would be strengthened by including specific experimental designs to directly verify the model's predictions.
  4. The limitations of the model are not adequately addressed. For example, the assumption of uniform water activity throughout the system may not hold in complex microbial cultures. A thorough analysis of the model's limitations and suggestions for future improvements should be included in the discussion.
  5. The quality of Figures 2 and 6 is subpar, and they lack sufficient captions. Enhancing the clarity of these figures and providing detailed captions would improve the manuscript.
  1. An example of the required validation appears in the new version. It concerns an IC trace from a culture of helveticus reproduced according to the thermodynamic description and use of the model of the virtual culture.
  2. and 3 The discussion about LTEE has been substantially extended, including also a proposal for an experimental design.
  3. The discussion about the uniformity of water activity now mentions the cases of non uniform distribution that are the subject of a previous work by the author.
  4. Figure 2 and 3 have been replaced, Figure 6 has beenimproved, and a new Figure 7 has been included.

Reviewer 2 Report

Comments and Suggestions for Authors

This article uses thermodynamics to explain how microbes grow and evolve, without relying on complex molecular models. It focuses on energy flow, entropy, and osmotic balance to describe growth and decay. The key idea is that both short- and long-term changes in microbes can be understood through these energy principles.

Please address the following: 

Is it possible to support thermodynamic claims with quantitative experimental data (even from literature) e.g., heat measurements, OD curves, metabolite concentration changes, or entropy calculations.

Validate predictions with real microbial growth datasets, including both batch and chemostat systems.

Link thermodynamic variables (like ∆G and entropy) with actual biochemical processes, such as ATP generation, enzyme activity, or nutrient uptake, to bridge the gap between theory and practice.

Discuss how this model might extend to other systems, and what constraints those environments impose on thermodynamic parameters.

what new insights the thermodynamic approach offers.

Author Response

Is it possible to support thermodynamic claims with quantitative experimental data (even from literature) e.g., heat measurements, OD curves, metabolite concentration changes, or entropy calculations.

Validate predictions with real microbial growth datasets, including both batch and chemostat systems.

Link thermodynamic variables (like ∆G and entropy) with actual biochemical processes, such as ATP generation, enzyme activity, or nutrient uptake, to bridge the gap between theory and practice.

Discuss how this model might extend to other systems, and what constraints those environments impose on thermodynamic parameters.

what new insights the thermodynamic approach offers.

  1. Many examples of experimental data reported in the literature are quoted throughout the text to support the proposed thermodynamic relationships
  2. The validation with real microbial cultures has been the subject of previous papers by the author. The model does not consider chemostatic cultures, which exchange matter with the external environment. Nonetheless, a proposal for an experimental design of chemostatic culture aimed at improving the current practice of LTEE has been included in the final section of the paper.

3.and 4               The links with some molecular character of the biochemical panoply underlying the microbial growth has been the subject of many papers, which are duly quoted throughout the manuscript. Most of them concern Thermochemistry, namely the first, rather than the second principle of Thermodynamics and the Thermodynamics of Irreversible Processes, which deal with the direction and the rate of the evolution of systems. Thermodynamics describes systems through their macroscopic properties that undergo changes because of physical constraints. Molecular details could be probes for tentative checks, but can unfortunately produce misleading evidences. These can indeed apply just to the singular detail considered and depend on the effect that physical (temperature, pH, water activity), chemical (kind and amount of substrate, molecular mass of cosolutes) and biological (starting density of the microbial population, presence of other microbes, adverse substances, etc.) conditions can produce. One should account for the whole body of molecular details to get a reliable perspective. This is possible only for very simple systems, like the ideal gas, while is almost impossible for a complex system, like a microbial cells. This is a main reason for the mess of discrepancies found in the literature.

Round 2

Reviewer 1 Report

Comments and Suggestions for Authors

I suggest that the manuscript be published in its current form.